# Grapevine Leafroll-Associated Virus 3 Replication in Grapevine Hosts Changes through the Dormancy Stage

**DOI:** 10.3390/plants11233250

**Published:** 2022-11-26

**Authors:** Mate Čarija, Silvija Černi, Darija Stupin-Polančec, Tomislav Radić, Emanuel Gaši, Katarina Hančević

**Affiliations:** 1Institute for Adriatic Crops, 21000 Split, Croatia; 2Department of Biology, Faculty of Science, University of Zagreb, 10000 Zagreb, Croatia; 3Selvita Ltd., 10000 Zagreb, Croatia

**Keywords:** virus replication, grapevine viruses, qPCR, relative quantification, grapevine leafroll disease symptoms

## Abstract

Grapevine leafroll-associated virus 3 (GLRaV-3) is a graft-transmissible virus present in every viticultural region of the world and poses a large threat to grapevine production. Frequent coinfections with other viruses, the large number of grapevine varieties, the complexity of processes involved in plant response to virus infection, and the lack of studies on GLRaV-3 replication limit our knowledge of GLRaV-3 damaging effects and their background. In this study, five different inocula, one containing GLRaV-3 and others containing GLRaV-3 in combination with different grapevine viruses were green grafted to 52 different grapevine plants of four varieties to analyze the influence of the phenological stage and virus composition on GLRaV-3 replication. Relative concentration analysis by quantitative PCR conducted over a 16-month period revealed that other viruses as well as plant stage had a significant effect on GLRaV-3 replication and symptoms expression. The replication was most pronounced in the deep dormancy stage at the beginning of the infection, and the least at the exit of the dormancy stage. This study brings new insight into GLRaV-3 replication and discusses about viral interactions in one of the most economically important perennial plants, the grapevine.

## 1. Introduction

Grapevine, along with other members of the *Vitis* genus host more than 80 virus species [1], which is attributed to a long history of domestication, lack of resistance sources within the *Vitis* genome, and germplasm exchange on a global level [2]. RNA viruses are the most common viruses infecting grapevine, with DNA viruses just recently being described [3]. Once they infect the host, grapevine viruses reside in them permanently and the plant uses many strategies to counteract them, RNA silencing being the most effective one [4]. Symptom development is a direct consequence of multiple interactions between factors such as the host and its capability to respond to virus infection, co-infecting viruses, environmental conditions, and vegetative season [5]. Within virus species described in grapevine, only some of them according to the accompanied damaging symptoms are considered especially harmful to grapevine performance and are included in certification programs of planting material [6]. One of them is grapevine leafroll-associated virus 3 (GLRaV-3, genus *Ampelovirus*, [7]), the main causal agent of grapevine leafroll disease (GLD) [8]. It is known to reduce plant vigor and longevity with a possible severe impact on fruit yield and quality [9]. Besides the damaging effect, GLRaV-3 poses a significant global concern for its wide presence and easy spreading [8].

GLRaV-3 is a graft transmissible virus that spreads by using infected propagation material and, within the vineyard blocks, by mealybugs vector transmission [8,10]. Replication of GLRaV-3 follows the replication strategy characteristic to all monopartite members of the *Closteroviridae* family where replication gene block (RGB) is translated directly from the capped virion RNA with a +1 frameshift translation for RdRp [11]. Genome replication is mediated by a complementary negative RNA strand to serve as a template for subsequent synthesis of genomic plus-strand RNAs as well as sgRNAs [12]. Most of the information obtained on GLRaV-3 replication and its potential impact on host plants are those from using methods such as quantitative Polymerase Chain Reaction (qPCR). Several assays have been developed so far for this purpose using either probe-based or SYBR green-based qPCR reactions [13,14,15] aiming at specific parts of the viral genome and measuring its positive-strand RNA synthesis. This approach allows for the quantification of target viral RNA rapidly and accurately, even if its concentration is low [14,15,16].

Frequent coinfections of GLRaV-3 with other viruses occur naturally in grapevine hosts, mostly with members of the genus *Vitivirus, Closterovirus, Maculavirus,* and *Nepovirus* [17,18,19]. New emerging viruses, such as *Trichovirus* grapevine pinot gris virus, also appear frequently in mixed infections [17], further adding to the confusion in elucidating interactions between viruses [18,20]. Those natural virus inocula limit the possibility of comparing virus dynamics in a single infection versus its dynamics when present in coinfection with other viruses [21]. So far, the relative virus concentration of *Vitivirus* genus members was found to be higher in plants coinfected with leafroll viruses but the incidence and relative virus concentration of grapevine leafroll viruses appeared to be unaltered by *Vitivirus* co-infection [20].

To our knowledge, there is limited information on complex interactions of GLRaV-3 with other viruses in mixed infections [16]. The aim of this study was to analyze the replication efficiency of GLRaV-3 in plants in single infection with GLRaV-3, or with GLRaV-3 in coinfection with other viruses. In total, 52 grapevine plants of four varieties were grown under greenhouse conditions and inoculated with five different inocula, each containing GLRaV-3. In three time points over a 16-month period, the effect of virus composition in infected plants as well as the effect of the plant’s phenological stages on the relative concentration of GLRaV-3 and other grapevine viruses were determined. A symptom assessment was conducted to study the impact of individual inoculum on four different grapevine varieties used in the study.

## 2. Results

### 2.1. Sanitary Status of Imported Cuttings and Inoculation Success

Randomly selected imported cuttings prior to inoculation all tested negative for the presence of 10 tested viruses as determined by PCR methods.

All scions from the donor plants that were green-grafted successfully developed new shoots within 15 days. In total, all 52 plants continued with their seasonal growth, and no plants were lost during the green grafting procedure. One Cabernet Franc treatment (Table 1) grafted with inoculum Z later declined and was lost for unknown reasons and excluded from the experiment. Petiole samples obtained from grafted plants three months after inoculation all tested GLRaV-3 positive by ELISA. By testing the GLRaV-3 transmissions, we could also assess the transmission of other viruses in mixed inocula. PCR confirmed the results obtained by ELISA for GLRaV-3, but also confirmed the transmission of other six viruses (GVA, GPGV, GRSPaV, GFkV, GLRaV-1 and GLRaV-3) in mixed inocula as listed in Table 1.

### 2.2. Reference Gene Selection

Ct values obtained for all experimental plants from the first sampling period (five months after inoculation) were used for calculating the most stable host reference gene, out of three candidates: actin, α-tubulin, and GAPDH. The analysis performed using Normfinder and geNorm proved that actin and α-tubulin displayed the most stable expression (Table 2). The expression of GAPDH showed to be less stable in comparison to other evaluated candidates and was excluded from further calculations.

For the Normfinder software stability values closer to 0 were considered to be of the reference genes with the most stable expression. geNorm stability value (M) is considered to be the most stable for those reference gene candidates whose value is M < 1.5.

Actin and α-tubulin genes were further evaluated for the expression stability over time in different phenological phases of plants by calculating the mean difference of Ct values. In the first sampling period (five months after inoculation) the mean difference in their Ct values was 2.1 ± 0.1, while in the second and the third (eight and 16 months after inoculation) it was 2.8 ± 0.15.

### 2.3. GLRaV-3 Load in Infected Plants over Time

When comparing Ct values obtained for two different genomic regions of GLRaV-3 (ORF1a and CP), no significant differences were observed by t-test. Results on relative GLRaV-3 concentration were displayed in relation to ORF1a genomic region, but the same results were obtained for the CP region that is expressed via sgRNA (not displayed here). The first step was to compare relative concentrations of GLRaV-3 in different grapevine varieties and no significant differences were observed when comparing individual varieties over time in all three sampling periods (five, eight, and 16 months post-inoculation). Since there were no significant differences between grapevine varieties all results were displayed on a *Vitis* level, regardless of variety status, but according to different time points and inoculum type.

Overall, the relative concentration of GLRaV-3 in all grapevine plants, regardless of variety status, in three time points detected the highest GLRaV-3 concentration in the first sampling period (five months post inoculation). The GLRaV-3 concentration measured in the other two time points (eight and 16 months post-inoculation) differed significantly from the first one and was significantly lower (Figure 1).

To check if the same effect applies to GLRaV-3 replication in plants infected only with GLRaV-3, we have tested the relative concentration of GLRaV-3 in all grapevine plants inoculated with GLRaV-3 only (inoculum II). Significant differences were observed for different sampling periods (Figure 2). In this case, the relative GLRaV-3 concentration in the first and the third sampling period was significantly higher than in the second one (Figure 2).

Analyzing relative GLRaV-3 concentration in relation to inoculum composition, infected plants did not show significant differences in GLRaV-3 replication in the first two sampling periods (five and eight months after inoculation). Only in the third time point (16 months after inoculation) comparison between plants infected with different inocula showed a significant difference for plants infected with inoculum II (only GLRaV-3) and the inoculum X (GLRaV-3 in coinfection with GVA, GPGV, and GRSPaV) (Figure 3). Plants infected with only GLRaV-3 had the highest relative virus concentration and those inoculated with combination X had the lowest.

### 2.4. Relative Concentration of Other Viruses

As in the case of GLRaV-3, differences in relative concentrations of other particular viruses in plants infected with inocula containing multiple viruses were observed in three time points and in relation to inoculum composition. Regarding the relative concentration of GVA, the first and the third sampling period differed significantly (Figure 4A). When observing the relative concentration of GVA between different inocula, inoculum Y (containing also GLRaV-3, GLRaV-1, GRSPaV, and GPGV) differed significantly from others in all three sampling periods. Results are here shown for the first sampling period (Figure 4B). but the same values were measured in the other two time points.

Relative concentrations of GPGV and GFkV were the highest in the first sampling period (five months after inoculation) and differed significantly with the second sampling period (eight months after inoculation) in both cases (Figure 5). In the plants inoculated with virus combination X, the relative concentration of GPGV was also the highest in the first sampling period and significantly higher than in the second and third sampling period (not shown).

GLRaV-1 and GLRaV-2, each present only per one inoculum (Table 1) displayed no significant changes in different sampling periods. The same referred to GRSPaV, which also displayed no changes among different inocula.

### 2.5. Symptom Expression of Grapevine Leafroll Disease

Symptom expression was evaluated for two consecutive years and ranked as asymptomatic, mild, moderate, and severe (Figure 6). In the first year, no symptoms of grapevine leafroll disease were noted. In the second year, symptoms of leaf reddening and downrolling were observed and classified for each variety individually depending on the inoculum used (Table 3). No correlation between symptom intensity and relative concentration of GLRaV-3 was established. The most expressed symptoms were observed in plants inoculated with Y combination harboring two GLD causal agents (GLRaV-1 and GLRaV-3; Table 3). Standard indicators (Cabernet Franc, Merlot, and Pinot Noir) displayed variable reactions to particular inoculum while Tribidrag displayed no symptoms, except when being infected with inoculum Y.

## 3. Discussion

The dynamics of GLRaV-3 manifest via its movement and replication, which are important for its infectivity. In the grapevine growing season of the northern hemisphere, GLRaV-3 in field-grown plants replicates and spreads quickly from the trunk to new growing shoots and leaves, but from September, GLRaV-3 quantitative concentration decreases [15], and GLRaV-3 is retreated in the phloem tissue of the trunk and in roots [15]. For this study, all phloem samples were taken from the basal part of the plant which is commonly used for the purposes of virus detection in the dormancy season of plants [22]. The dormancy stage is a key feature of perennial plants where plant physiological activities are reduced to a minimum as the growth itself is suspended [23]. To analyze the GLRaV-3 concentration during the dormancy period first and third samplings in the one-year interval were performed at the beginning of the dormancy phase (five and 16 months post inoculation) when the plant’s growth is deeply suspended. The second sampling was performed at the end of the dormancy period when the plant begins to exit the dormancy phase (eight months post inoculation). As proven by analyzing plants infected with GLRaV-3 only (inoculum II), the relative GLRaV-3 concentration differed significantly between different periods of dormancy, being significantly higher in the deep dormancy period (first and third samplings). This indicates different GLRaV-3 replication efficiency through the dormancy phase. The second sampling, performed at the end of the deep dormancy period was also characterized by significantly lower virus replication compared to the first one. It was the most obvious in plants infected only with GLRaV-3 (inoculum II). This finding is confirmed in our study by observing the cumulative effects of all plants inoculated with different combinations of viruses, where GLRaV-3 measured a significantly higher concentration in the first sampling point in comparison to the other two sampling points. Based on these results, we can assume that a prolonged period of viral infection and/or suspended growth had a negative effect on the replication of GLRaV-3. In our study, besides the GLRaV-3, the same effect was observed when measuring relative concentrations of GPGV and GFkV.

The inoculum had mostly no influence on GLRaV-3 replication, at least until the last sampling point that occurred 16 months post-inoculation. In that period the relative concentration of the GLRaV-3 differed significantly between plants infected with GLRaV-3 alone (inoculum II) and those infected with inoculum X (GLRaV-3 in combination with GVA, GPGV, and GRSPaV; Figure 3). The lower GLRaV-3 concentration observed in plants grafted with inoculum X was probably due to the effect of mixed infections, as suggested by Tsai et al. [15] who associated the lower concentration of GLRaV-3 with the presence of GVB and GFkV Results of our study indicate that particular virus combination within the X inoculum has a negative effect on GLRaV-3 replication and that the duration of infection may have influenced the GLRaV-3 replication. Since the same viruses as in the X inoculum were also present in Y and Q inocula with no significant effect on GLRaV-3 replication, according to our results we can assume that the effect of a particular virus combination prevails over the effect of an individual virus. Furthermore, inocula Y and Q contained other causal GLD agents (*Ampelovirus* GLRaV-1 and *Closterovirus* GLRaV-2 respectively) and no evidence of synergistic or antagonistic relationship was found in terms of relative GLRaV-3 concentration. This is in accordance with the previous results of Velasco et al. [24] indicating that *Ampeloviruses* do not interact with each other on this level, but it is not excluded that the causal agents of GLD may interact with other viruses present in the inoculum. Although some researchers detected dependency of GVA and GLRaV-3 for GVA vector transmission efficiency [25], our study did not show any significant interactions in terms of GVA and GLRaV-3 replication efficiency following plants inoculated with inoculum Z (GLRaV-3 and GVA) in different dormancy stages.

The relative concentration of GVA differed significantly amongst mixed inocula where inoculum Y had a significantly lower concentration compared to all other inocula in all three sampling periods. Since the only difference between inocula Y and X is the presence of GLRaV-1 in the inoculum Y, this most likely further adds to a prevailing effect of virus combination over the effect of an individual virus. This is the opposite of what was previously reported that *Vitiviruses* have a positive correlation with *Ampeloviruses* [20] in terms of relative concentrations and enhanced vector transmission. Probably the effect of coinfection of two *Ampeloviruses* along with GPGV and GRSPaV had a negative effect on the replication of GVA in this particular inoculum used. Also, we cannot exclude the possible effect of other viruses that might have been present in the inocula but were not tested, since donor plants and later, grafted plants were checked for only 10 economically important viruses out of many more described. This potential problem was resolved by using the same donor plant source for every inoculum type.

Proper selection of the host reference genes is the main prerequisite for the relative quantification of genes of interest and reference genes ought to be stably expressed in different experimental conditions and in different tissues [26]. For this purpose, in grapevine, several candidates have been evaluated in studies under specific conditions with different genes of interest being observed [27,28]. In our case, actin and α-tubulin were chosen as the best candidates in the case of multiple virus infections in different *V.vinifera* indicator plants. On the basis of differences in their Ct values, we evaluated their stability in expression over different phenostages with only the first one slightly differing from the other two. This could be due to stressful events that preceded this sampling point such as the transmission of the viruses into a healthy plant via green grafting. Overestimation of virus population titer is one of the main concerns of quantitative PCR analysis, especially in the case of RNA viruses that use sgRNA expression strategy during their life cycle [13,15]. In this study, no significant differences were found between the two genomic regions of GLRaV-3 (ORF1a and CP) which confirms the finding of Bester et al. [14] who obtained the same results. This was also reported by Bertazzon et al. [29] who used two primer sets for the quantification of GPGV (one for the RdRP domain and the other for CP) and did not observe any significant differences in their relative expressions.

Symptom expression of grapevine leafroll disease (GLD) varied among different inocula used and also between different varieties used in this experiment. Tribidrag, a Croatian indigenous variety known as Primitivo and Zinfandel, was also included in this study along with standard indicator varieties (Merlot, Cabernet Franc, and Pinot Noir; [30]). Even though Tribidrag generally did not display GLD symptoms, inoculum Y provoked the strongest reaction in his case, such as in the case of other varieties observed in this study (Table 3). Since the inoculum Y was composed of five different viruses, two of them (GLRaV-1 and GLRaV-3, causal agents of GLD) belonging to the genus *Ampelovirus*, it is not surprising that inoculum Y developed the most intense GLD symptoms. Even though GLRaV-3 is considered to be the main causal agent of GLD [8], the effect of another *Ampelovirus* adds to symptom severity in different indicator plants (Table 3) despite the fact that no correlation was established in terms of their relative expression. This also corresponds to previous work on the red variety Plavac Mali [19] where this type of coinfection leads to stable expression of GLD symptoms in plants grown in field conditions. One intriguing observation of this study is that green-grafted plants did not display any GLD symptoms in the post grafting period, even though the virus titer was detectable as shown by ELISA and PCR. This is probably due to the short period after grafting to environmentally induced dormancy in the greenhouse, since it is well-known that GLD symptoms are better expressed in cooler climates [31]. In a very rare situation, the greenhouse conditions could be a factor preventing symptom expression, as shown by Constable et al. [32] who reported an almost total absence of GLD symptoms in Cabernet Franc indicator plants over a 3-year period in greenhouse conditions.

Even though grapevine plants were in the dormancy phase, with plant activities reduced to a minimum, the replication of viruses occurred following their individual replication pattern. GLRaV-3 replication in single-infected plants was strongly influenced by the prolonged period of virus infection and/or suspended plant growth which might have a negative influence on virus replication. GFkV and GPGV, as constituents in mixed infections along with GLRaV-3, followed the same trend: deep dormancy had a positive effect on virus replication as opposed to the end of dormancy, when virus replication was reduced significantly. This pattern was not observed in plants infected with GLRaV-3 in combination with other viruses, indicating that virus composition and the duration of infection might influence GLRaV-3 replication.

## 4. Materials and Methods

### 4.1. Plant Material and Virus Inoculation

*Vitis vinifera* varieties (Merlot, Pinot Noir, Cabernet Franc, and Tribidrag) were produced by rooting one-year-old virus-free grapevine cuttings. The dormant cuttings of Merlot, Pinot Noir, and Cabernet were imported as certified plant material from the Institut für Rebenzüchtung (Hochschule Geisenheim University, Geisenheim, Germany) and Tribidrag was imported as initial material from the Foundation Plant Service (University of California, Davis, CA, USA). Randomly selected cuttings (10% of the total number) were chosen for confirmation of sanitary status as will be described in the following sections. All other cuttings were rehydrated, and treated with fungicides against *Botrytis cinerea* and with a 2000 ppm solution of indole-3-butyric acid (IBA; Sigma Aldrich, Darmstadt, Germany) to induce rooting. Immediately after immersion of the basal part in the hormone solution, the cuttings were put into a mixture of perlite (Grand Quality 2–6 mm) and peat (Substrate stock with perlite, Gramoflor sondermischung; Lirstal, Germany) in a ratio of 3:1 to achieve optimal moisture retention with good aeration important for callus development and root growth. Rooting was performed on a heating table, and cuttings were irrigated every day/every two days to prevent drying out before root formation.

Four weeks after, plants with developed roots were transferred to 6 L pots filled with a mixture of soil (brown soil): peat (Brill type 5): perlite (Agrilit 3, Perlite espansa; Perlite Italiana, Milano, Italy) in a ratio of 1:1:1 and with 1/3 of the total mass fraction of quartz sand (Lasselberger –Knauf, Đurđevac, Croatia). Plants were growing in the greenhouse (Schwarzmann), under the natural light, following a temperature range from 18 to 35 °C and relative humidity between 30 and 80% depending on the season and irrigation regime. Plants were irrigated with 1/4 of Hoagland solution [33] once a week from the bud bursting stage until the summer season when they were irrigated with a nutrient solution once every two weeks and regularly treated against pests.

In the early summer of the second vegetation year, each grapevine variety was inoculated with five virus inocula containing GLRaV-3 and different viruses in combinations with GLRaV-3 (Table 1). Grapevine donor plants used for inoculation were previously proven to be infected with particular viral composition by Hančević et al. [17] as the most frequent virus combinations in grapevine varieties from the Mediterranean part of Croatia. Different indicators were inoculated using the same inoculum source of the same donor plant. Grafting was performed by the green grafting technique [34]. Herbaceous cuttings 8 to 10 cm long and 1.5 to 2.5 mm in diameter were collected from the donor plants and the slanting cut was performed on every donor cutting to fit the slant in the indicator plant and tied with plastic tape (Figure 7). In total 52 plants were grafted. Grafted plants were intensively watered and a single use of nitrogen-based fertilizer (ammonium nitrate, 1 g/L) was applied to induce new vegetative growth.

### 4.2. Symptom Assessment

Grapevine leafroll disease symptoms were assessed for two consecutive years throughout the summer period and into the early autumn. Individual plants were classified as asymptomatic, with mild (occasional interveinal reddening associated with GLD), moderate (interveinal reddening with leaf downroll that affected up to half of the plant leaves), and severe symptoms (extensive interveinal reddening with leaf downroll with more than half of the leaves affected).

### 4.3. Virus Detection in Imported Cuttings and Inoculated Grapevine Plants

Preliminary confirmation of the virus transmission on grafted plants was carried out 3 months after inoculation (in autumn) by ELISA test targeting GLRaV-3. DAS-ELISA [35] was performed on petioles collected from each inoculated plant using a GLRaV-3 commercial kit (Agritest, Valenzano, Italy). After grinding 200 mg of plant tissue in liquid nitrogen, samples were diluted in 2 mL of extraction buffer and prepared according to the manufacturer’s instructions. The samples were incubated with capture antibody at 4 °C overnight, followed by incubation with conjugate antibody the next day for 2 h at 37 °C. Two and three hours after adding substrate p-nitrophenyl phosphate, absorbance was recorded at 405 nm using a microplate reader (Multiscan, Thermofisher; Waltham, MA, USA). Absorbance values 2.5 times greater than the mean absorbance value of the negative control were considered positive to virus presence.

To confirm the sanitary status of imported cuttings and the transmission of all viruses from inocula to indicator plants, PCR-detection was performed from phloem tissue collected from the basal part of the plants.

The total RNA was extracted from 250 mg of cortical scrapings with RNeasy Plant Mini Kit (Qiagen, Hilden, Germany), applying an improved RNA extraction procedure modified by Mackenzie et al. [36]. Reverse transcription was performed using 200 units of MMLV reverse transcriptase (Invitrogen, Waltham, MA, USA), 100 units of RNase inhibitor (Invitrogen, Waltham, MA, USA), 0.5 mM dNTPs and 5 µM random nonamers (Sigma Aldrich, St Louis, MO, USA) using 250 ng of RNA template. The reaction mixture was incubated for 10 min at 25 °C and 60 min at 37 °C followed by 15 min at 70 °C.

Multiplex PCR was performed as described by Gambino and Grimbaudo [37] for GLRaV-1, GLRaV-2, GLRaV-3, GFLV, ArMV, GFkV, GVA, GVB, and GRSPaV. Transmission of GPGV was tested by an additional PCR as described by Saldarelli et al. [38]. As indicators of RNA quality and RT-PCR effectiveness, primers for *Vitis* 18S rRNA were used and for negative control RNase, free water was added instead of a cDNA template. Reaction products were analyzed by agarose gel electrophoresis ascertaining the following amplicon sizes: 18S rRNA (844 bp), GPGV (588 bp), GLRaV-2 (543 bp), GLRaV-3 (336 bp), GVA (272 bp), GLRAV-1 (232 bp), GFkV (179 bp) and GRSPaV (155 bp).

### 4.4. Viral Load Quantification in Grapevine Plants

Canes from grapevine indicator plants were pruned and sampled during the dormancy stage in three time points: 5 months (BBCH 00, [39]); 8 months (BBCH 01), and 16 months post-inoculation (BBCH 00). Total RNA from the phloem tissue was extracted as described previously. Sampled tissue of each variety harboring the same inoculum was pooled and treated as one. In total 19 samples were analyzed by qPCR.

### 4.5. Primer Selection and Reaction Conditions

Primers chosen for reference gene candidates were those encoding for actin, α-tubulin, and GAPDH as described by Reid et al. [27]. For GLRaV-3 quantification, two primer pairs encoding for ORF 1a and CP virus regions were selected as described by Bester et al. [14]. Primers for quantification of GVA, GVB, GLRaV-2, and GFkV were used as described by Faggioli et al. [40], for GLRaV-1 as described by Gambino and Gribaudo [37] and for GPGV and GRSPaV quantification as described by Bertazzon et al. [29]. All primers were evaluated using serial dilutions from a pooled cDNA sample and all the primers met the criteria for qPCR analysis as described by Thornton and Basu [41]. Sequences and concentrations of all primers used in qPCR mix are listed in Table 4.

RT-qPCR mix for all reactions contained 1XSYBR Green (Applied Bio-systems, Carlsbad, CA), forward and reverse primers as described by the authors [14,27,29,37,40], and 1 µL of cDNA template. Cycler conditions used were as described by Bester et al. [14] and melt curve analysis was performed to verify the primer specificity. All reactions were performed in triplicate and their average value was used for further calculations (Average values shown in Appendix A).

### 4.6. Reference Gene Candidate Selection and Relative Concentration of Virus mRNA

Reference gene candidates for relative virus quantification were evaluated using geNorm [42] and Normfinder [43]. The relative concentration of viral RNA was determined by normalizing the virus Ct value to the geometric mean of selected reference genes’ CT value using the ΔCt method [44]. The difference in expression among varieties was calculated by normalizing the ΔCt values of individual treatments to the Ct value of the control sample for each variety and finally expressed as ΔΔCt [44]. The control sample used was inoculum II, harboring only GLRaV-3.

### 4.7. Statistical Analysis

Statistical analysis (ANOVA) was performed using TIBCO Statistica™ (Version: 14.0.1; [45]) after the logarithmic transformation of the data. A comparison of the ΔCt values of the two genomic regions of GLRaV-3 (ORF1a and CP) was performed by t-test. For comparing the differences between different phenological stages repeated measures ANOVA was used. For multiple comparison of different varieties and inocula in the same sampling period, one-way ANOVA was used. Bonferroni post hoc test was conducted and statistically significant values (*p* < 0.05) were taken into consideration.

## Figures and Tables

**Figure 1 plants-11-03250-f001:**
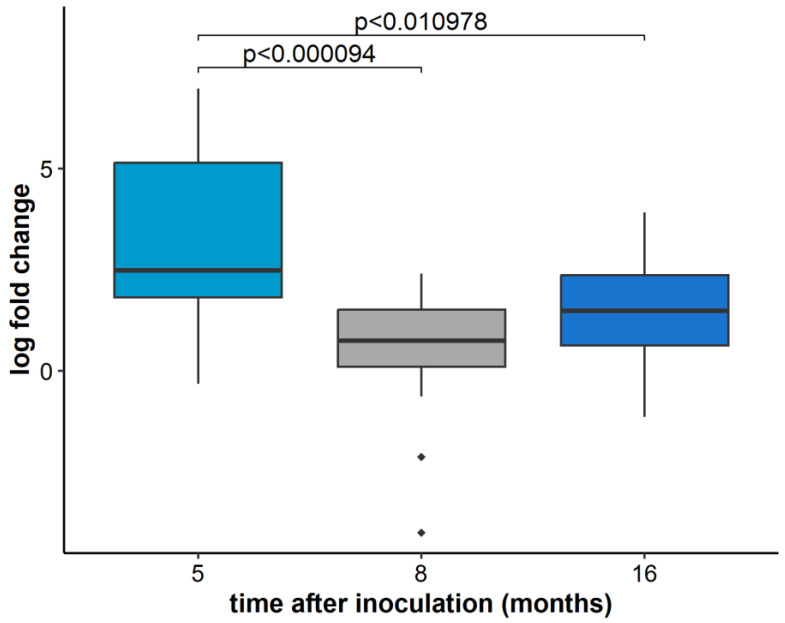
The relative concentration of GLRaV-3 based on ORF1a analysis in five, eight, and 16 months after inoculation, calculated as an average value for all inoculated plants of all varieties. Results are displayed as logarithmic ratios of relative concentration. ANOVA test was performed along with the Bonferroni post hoc test and significantly different values (*p* < 0.05) are marked above individual box plots. Outlier measurements that are 1.5 times over the upper/lower quartile of the dataset of an individual boxplot are marked with ◆.

**Figure 2 plants-11-03250-f002:**
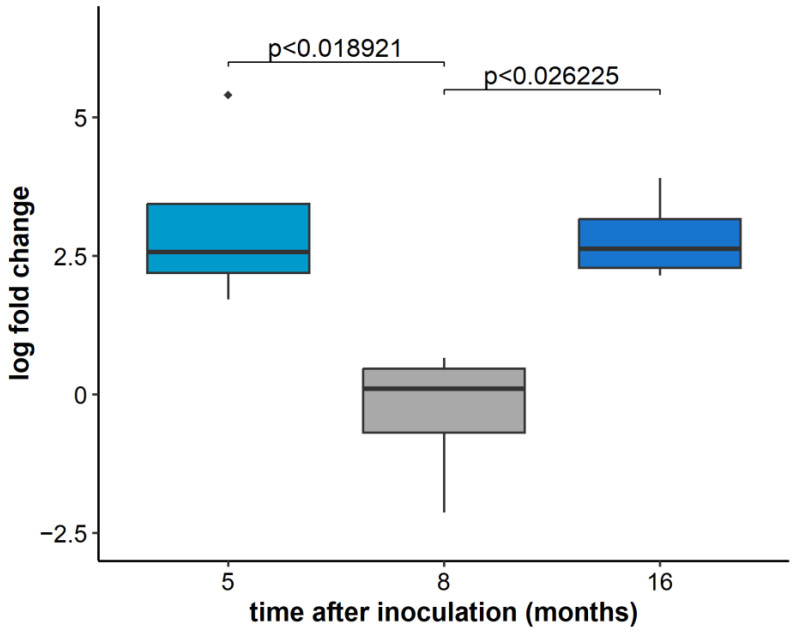
The relative concentration of GLRaV-3 based on ORF1a analysis five, eight months, and 16 months after inoculation observed in plants of all varieties harboring GLRaV-3 only (inoculum II). Results are displayed as logarithmic ratios of relative concentration. ANOVA test was performed along with the Bonferroni post hoc test and significantly different values (*p* < 0.05) are marked above individual box plots. Outlier measurements that are 1.5 times over the upper/lower quartile of the dataset of an individual boxplot are marked with ◆.

**Figure 3 plants-11-03250-f003:**
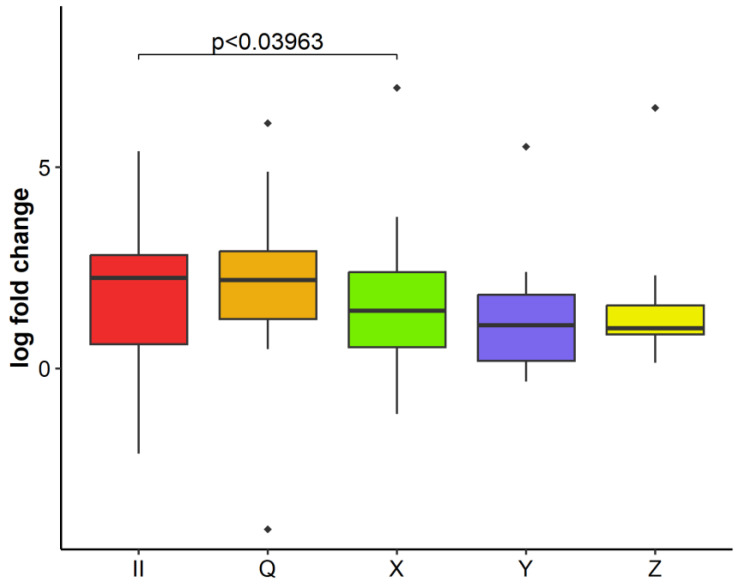
Multiple comparisons of relative concentration of GLRaV-3 ORF1a observed 16 months after inoculation of all plants of all varieties infected with the same inoculum (II, Q, X, Y or Z). Results are displayed as logarithmic ratios of relative concentration. ANOVA test was performed along with Dunnett’s post-hoc test and significantly different values (*p* < 0.05) are marked over individual box plots. All concentrations were compared with the one of inoculum II (control) being the one harboring only GLRaV-3. Outlier measurements that are 1.5 times over the upper/lower quartile of the dataset of an individual boxplot are marked with ◆.

**Figure 4 plants-11-03250-f004:**
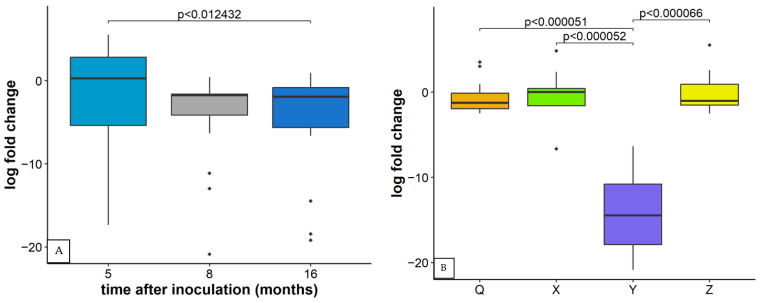
(**A**) Relative concentration of GVA based on CP region five, eight, and 16 months after inoculation observed in inoculated plants of all varieties. Results are displayed as a logarithmic ratio of relative concentration. ANOVA test was performed along with Bonferroni posthoc test and significantly different values (*p* < 0.05) are marked over individual box plots. (**B**) Multiple comparisons of relative concentration of GVA based on CP region five months after inoculation in plants of all varieties harboring the same inoculum. Results are displayed as a logarithmic ratio of relative concentration. ANOVA test was performed along with the Bonferroni post hoc test and significantly different values (*p* < 0.05) are marked over individual box plots. In both cases, Outlier measurements that are 1.5 times over the upper/lower quartile of the dataset of an individual boxplot are marked with ◆.

**Figure 5 plants-11-03250-f005:**
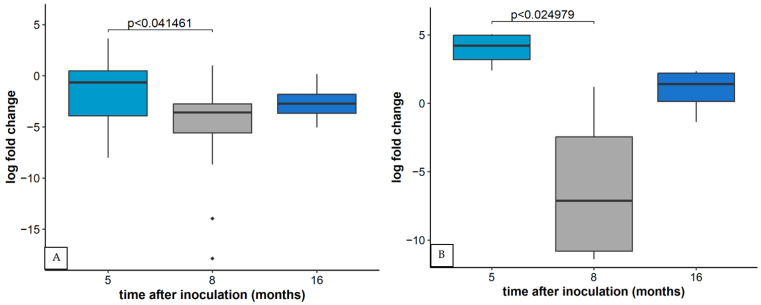
(**A**) The relative concentration of GPGV based on CP region and (**B**) GFkV based on CP region five, eight, and 16 months after inoculation observed in inoculated plants of all varieties. Results are displayed as logarithmic ratios of relative concentration. ANOVA test was performed along with the Bonferroni post hoc test and significantly different values (*p* < 0.05) are marked over individual box plots. Outlier measurements that are 1.5 times over the upper/lower quartile of the dataset of an individual boxplot are marked with ◆.

**Figure 6 plants-11-03250-f006:**
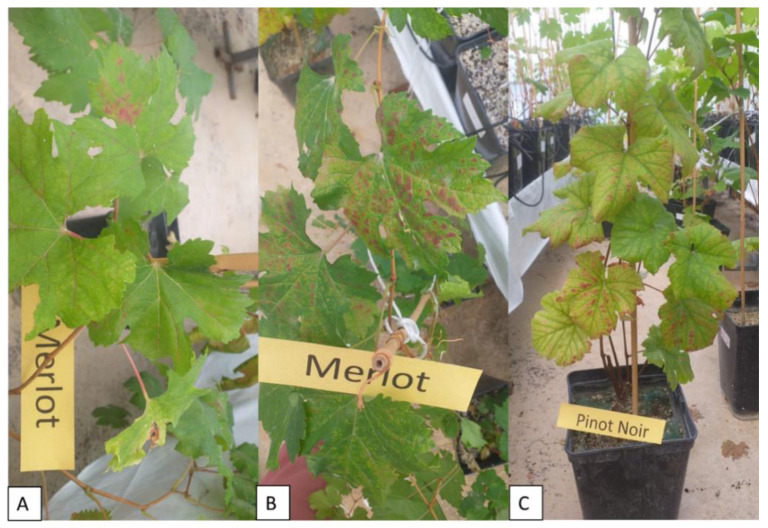
Expression of symptoms, categorized as (**A**) mild, (**B**) moderate, and (**C**) severe in grapevine varieties under the greenhouse conditions.

**Figure 7 plants-11-03250-f007:**
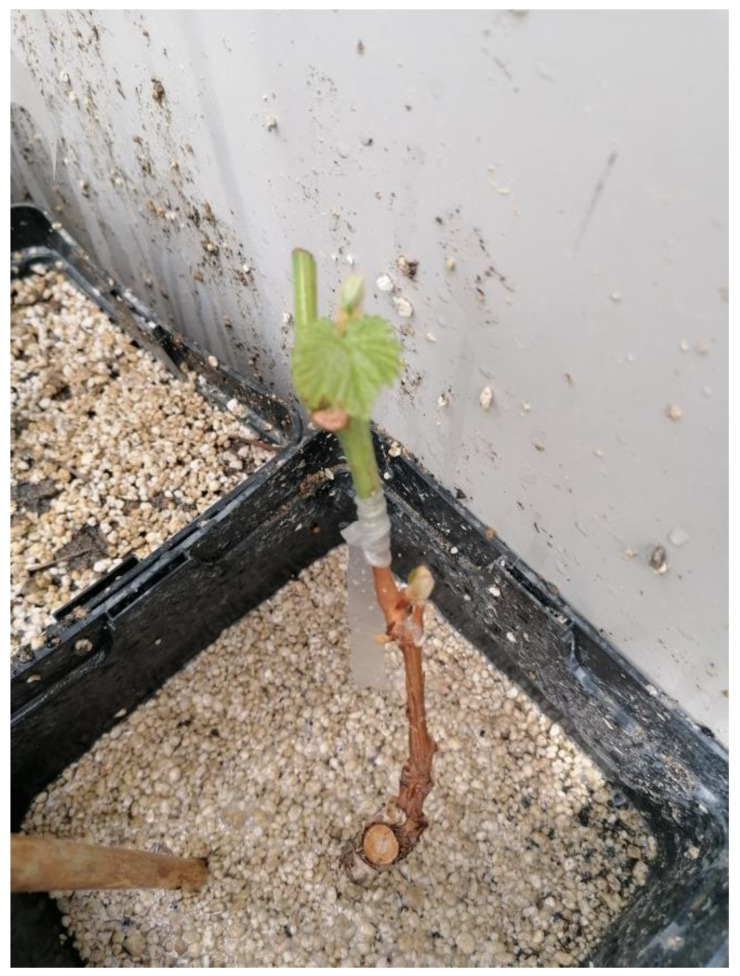
Green grafted *Vitis vinifera* plant for the purpose of virus transmission.

**Table 1 plants-11-03250-t001:** Viral composition of different inocula along with the number of plants grafted per each grapevine variety.

Name of the Inoculum	Viral Composition of the Inoculum	Number of Plants Grafted per Each Grapevine Variety			
		TR	PN	CF	M
II	GLRaV-3 *	6	2	3	1
X	GLRaV-3, GVA, GPGV, GRSPaV	5	2	1	2
Y	GLRaV-3, GVA, GLRaV-1, GPGV, GRSPaV	5	2	1	2
Z	GLRaV-3, GVA	5	2	/ **	2
Q	GLRaV-3, GVA, GLRaV-2, GFkV, GPGV, GRSPaV	5	3	1	2

* Abbreviations stand as listed: GLRaV-3- grapevine leafroll-associated virus 3, GVA- grapevine virus A, GPGV- grapevine pinot gris virus, GRSPaV- grapevine rupestris stem pitting associated virus, GLRaV-1- grapevine leafroll-associated virus 1, GLRaV-2- grapevine leafroll-associated virus 2, GFkV- grapevine fleck virus. Abbreviations for the grapevine varieties are as follows: TR- Tribidrag, PN- Pinot Noir, CF- Cabernet Franc and M- Merlot. ** Plants excluded from the experiment as they declined.

**Table 2 plants-11-03250-t002:** Reference gene validation results by Normfinder and geNorm software.

Reference Gene Candidate	Normfinder Stability Value	geNorm Stability Value
Actin	0.28	0.454
α-tubulin	0.14	0.454
GAPDH	0.70	1.336

**Table 3 plants-11-03250-t003:** Grapevine leafroll disease symptoms’ development in different grapevine indicators inoculated with different virus combinations (inocula).

Inoculum	Virus Composition	Indicator	GLD Symptom Development
II	GLRaV-3 *	TR	AS *, AS, AS, AS, AS
		M	MO
		PN	MO, AS
		CF	MI, MI, MI
X	GLRaV-3, GVA, GRSPaV, GPGV	TR	AS, AS, AS, AS, AS
		M	MI, S
		PN	AS, AS
		CF	AS
Y	GLRaV-3, GLRaV-1, GVA, GRSPaV, GPGV	TR	S, AS, S, MO, MO
		M	S, AS
		PN	AS, S
		CF	MI
Q	GLRaV-3, GLRaV-2, GVA, GFkV, GRSPaV, GPGV	TR	AS, AS, AS, AS, AS
		M	MO, MI
		PN	S, S
		CF	AS
Z	GLRaV-3, GVA	TR	AS, AS, AS, AS, AS
		M	MI, AS
		PN	MO, MI

* Abbreviations for individual viruses stand as listed: GLRaV-3- grapevine leafroll-associated virus 3, GVA- grapevine virus A, GPGV- grapevine pinot gris virus, GRSPaV- grapevine rupestris stem pitting associated virus, GLRaV-1- grapevine leafroll-associated virus 1, GLRaV-2- grapevine leafroll-associated virus 2, GFkV- grapevine fleck virus. Indicator plants are listed as follows: TR- Tribidrag, M- Merlot, PN- Pinot Noir, and CF- Cabernet Franc. Symptoms were classified as AS (asymptomatic), MI (mild), MO (moderate), and S (severe).

**Table 4 plants-11-03250-t004:** Primer targeting *Vitis vinifera* host genes (Vv) and grapevine virus genes.

Primer	Primer Sequence (5′–3′)	Primer Concentration
Vv actin	F: CTTGCATCCCTCAGCACCTT	0.4 µM
	R: TCCTGTGGACAATGGATGGA	
Vv tubulin	F: CAGCCAGATCTTCACGAGCTT	0.4 µM
	R: GTTCTCGCGCATTGACCATA	
Vv GAPDH *	F: TTCTCGTTGAGGGCTATTCCA	0.4 µM
	R:CCACAGACTTCATCGGTGACA	
GLRaV-3_ORF1a	F: GGGRACGGARAAGTGTTACC	0.4 µM
	R: TCCAAYTGGGTCATRCACAA	
GLRaV-3_CP	F: ATGAAYGARAARGTYATGGC	0.48 µM
	R: CTAAACGCYTGYTGYCTAG	
GLRaV-2	F: CAAATAGTTTCGGAGAGAGGAATG	0.33 µM
	R: GCGATACAAAAGTCAACGTAAGC	
GLRaV-1	F: TCTTTACCAACCCCGAGATGAA	0.33 µM
	R: GTGTCTGGTGACGTGCTAAACG	
GVA	F: GACAAATGGCACACTACG	0.33 µM
	R: AAGCCTGACCTAGTCATCTTGG	
GFkV	F: CTAGCTCTCGCTCTGACTCT	0.33 µM
	R: TCATACCACAGGAACTGGAA	
GRSPaV	F: AATAATTCCCCGATTTCAAGGC	0.33 µM
	R: AGGATTTAGCATRGAAAGGGAATAC	
GPGV	F: TGTCGATTCGTCAGGAGTTG	0.33 µM
	R: GGGTAAATTGTCTCCCCTGA	

* Abbreviations stand as listed: GAPDH- Glyceraldehyde 3-phosphate dehydrogenase, GLRaV-3_ORF1a- ORF1a genomic region of grapevine leafroll-associated virus 3, GLRaV-3_CP- Capsid protein genomic region of grapevine leafroll-associated virus 3, GLRaV-2- grapevine leafroll-associated virus 2, GLRaV-1- grapevine leafroll-associated virus 1, GVA- grapevine virus A, GFkV- grapevine fleck virus, GRSPaV- grapevine rupestris stem pitting associated virus and GPGV- grapevine pinot gris virus.

## Data Availability

The average Ct values of the results presented in this study is available in Appendix A.

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
