# Peer review of "Grapevine Leafroll-Associated Virus 3 Replication in Grapevine Hosts Changes through the Dormancy Stage"

_plants, 2022, doi:10.3390/plants11233250_

Round 1
Reviewer 1 Report
Dear authors,
I have reviewed your manuscript " GLRaV-3 replication in grapevine hosts changes through the dormancy stage" submitted for publication in Plants. The manuscript is interesting and presents a valuable collection of information on the influence of the phenological stage and virus composition on GLRaV-3 replication in different host combination. However, some points need to be improved and detailed before the manuscript is accepted for publication. Add the complete information on cutting propagation. Add a figure showing the grafting process; it is necessary to describe this procedure in detail. Some sentences require the addition of references. Authors need to describe in detail the conditions of the greenhouse (temperature, light, humidity) where the plants were grown, the complete DAS-ELISA and PCR procedures. I know you referenced articles for the primers; however, it is necessary to add this information here; I strongly recommend that you describe it in a table. Are the data presented in the figures the average of all cultivars? Please add this information to the figure caption.
It is necessary to describe all procedures in detail, otherwise others cannot repeat your study.
I have highlighted a number of suggestions for your consideration:
Line 1: Describe GLRaV-3
Line 4-6: Use the same theme fonts. Add the author’s email.
Line 8: Correspondence: Katarina.Hancevic@krs.hr, Tel +38521434435 8 (K.H.); silvija.cerni@biol.pmf.hr, Tel +38514898095 (S.C.)
Line 10: enter a space before the word “Grapevine”
Lines 27-28: Grapevine, along with other members of the Vitis genus, hosts over 80 virus species [1], which is attributed to
Line 38: Add a reference to support this sentence
Line 40: [7]. It is known
Lien 42: Add a reference to support this sentence
Line 58: Nepovirus [add references to support this sentence].
Line 66: As far as we know, there is limited information on complex interactions of GLRaV-3 in mixed infections [add references to support this sentence].
Line 68: in single infection with
Line 80: new shoots after xxx days/weeks
Lines 87-88: Table 2 and 3.
Line 94: Table 1
Lines 116-177: add the period
Line 118: add the periods
Line 119: (Figure 1). Please review this throughout the document.
Figures: Improve the resolution quality of the caption in the axis x and y. Are these data the average of all cultivars? Add this information to all figures.
Line 120: Time after inoculation (months). Same comment for figure 4A.
Line 168: “A” and “B” are offset in the figure
Lines 193 and 200: (Table 2). Please review this throughout the document.
Line 198: Have you noticed any differences among cultivars? Add this information.
Line 211: Add a figure showing the disease symptom
Please review the word phloem throughout the document. You wrote floem!
Lines 243-245: You used another theme fonts
Line 287: varied among different
Line 322: Mention that they are Vitis vinifera cultivars
Lines 323-324: Add the complete information on cutting propagation. Add the period of the year and stage of development of the plants that the cuttings were collected, herbaceous or woody cuttings, how long was the incubation in IBA? etc. You can also reference a manuscript.
Line 324: Describe IBA
Lines 327-330: regularly watered? Add complete information
Line 327: How long did it take to root the cuttings?
Line 330: Add the greenhouse conditions (temperature, humidity, light).
Line 330: Hoogland solution [add a reference]
Line 337: Add a figure showing this process. In what position in the plant? Describe in detail.
Lines 349-355: Add a figure showing the disease symptom
Lines 358-369: Describe in detail the DAS-ELISA and PCR protocols... including amount of plant tissue, season, etc.
Line 372: add the seasons. Were plants dormant? Or at what stage specifically?
Line 373: " Total RNA from the floem tissue was extracted as described previously.” You do not describe this earlier; it is necessary to describe all procedures in detail, otherwise others cannot repeat your study.
Line 421: add complete reference
Please carefully revise all these points.
Author Response
We thank the Reviewer 1. for reviewing our work and for all comments and suggestions that we find very beneficial for improving the manuscript quality.
Please see the attachment.
KInd regards,
Authors

Reviewer 2 Report
In this paper, quantitative PCR analysis of relative concentrations over 16 months found that specific virus combinations and different plant periods had significant effects on GLRaV-3 replication and symptom expression.The methods and results are logical and solid.In the discussion part, every result of the paper is fully analyzed and supposed. while some minor problems should be corrected or clarified. In my opinion, it can be accepted after minor revision.
In the symptom expression of grapevine leafroll disease, it is recommended to add pictures of the typical symptoms of the four symptom levels.In this way, the symptoms of different inocula in different varieties can be seen more clearly.
Minor points:
P.1 line 217:suggest changing" floem tissue"to"phloem tissue"
P.2 line 243-245:I recommended to use the same font as the body text.
P.3 line 362:same as P1.
P.4 line 373:same as P1.
P.5 line 389:If this is a subheading, I recommend using the same format as the other subheadings in the article.
Author Response
We are very thankful for reviewing the manuscript and for all comments. We welcomed all suggestion of the reviewer and thank for the her/his contribution for the manuscript improvement.
Please see the attachment.
KInd regards,
Authors

Round 2
Reviewer 1 Report
All recommendations were added, and questions clarified.
Author Response
Authors thank the reviewer for her/his valuable comments.